# Surface respiratory electromyography and dyspnea in acute heart failure patients

**Daniele Luiso**[1,2], **Jair A. Villanueva**[3], **Laia C. Belarte-Tornero**[1], **Aleix Fort**[1], **Zorba Blázquez-Bermejo**[1], **Sonia Ruiz**[1,4], **Ramon Farré**[3,5,6], **Jordi Rigau**[3,7], **Julio Martí-Almor**[1,2,4], **Núria Farré**[1,2,4]*

1 Department of Cardiology, Hospital del Mar, Barcelona, Spain, 2 Department of Medicine, Universitat Autònoma de Barcelona, Barcelona, Spain, 3 Unit of Biophysics and Bioengineering, School of Medicine and Health Sciences, Universitat de Barcelona, Barcelona, Spain, 4 Heart Diseases Biomedical Research Group (GREC), IMIM (Hospital del Mar Medical Research Institute), Barcelona, Spain, 5 CIBER of Respiratory Diseases (CIBERES), Madrid, Spain, 6 Institute for Biomedical Research August Pi Sunyer (IDIBAPS), Barcelona, Spain, 7 Research, Development, and Innovation Department, Sibel SAU, Barcelona, Spain

* NFarreLopez@parcdesalutmar.cat

**Data Availability Statement:** All relevant data are within the manuscript.

**Funding:** This work was supported by Sibel S.A.U. The funder provided support in the form of salaries

## Abstract

Introduction and Objectives: Dyspnea is the most common symptom among hospitalized patients with heart failure (HF) but besides dyspnea questionnaires (which reflect the subjective patient sensation and are not fully validated in HF) there are no measurable physiological variables providing objective assessment of dyspnea in a setting of acute HF patients. Studies performed in respiratory patients suggest that the measurement of electromyographic (EMG) activity of the respiratory muscles with surface electrodes correlates well with dyspnea. Our aim was to test the hypothesis that respiratory muscles EMG activity is a potential marker of dyspnea severity in acute HF patients. Methods: Prospective and descriptive pilot study carried out in 25 adult patients admitted for acute HF. Measurements were carried out with a cardio-respiratory portable polygraph including EMG surface electrodes for measuring the activity of main (diaphragm) and accessory (scalene and pectoralis minor) respiratory muscles. Dyspnea sensation was assessed by means of the Likert 5 questionnaire. Data were recorded during 3 min of spontaneous breathing and after breathing at maximum effort for several cycles for normalizing data. An index to quantify the activity of each respiratory muscle was computed. This assessment was carried out within the first 24 h of admission, and at day 2 and 5. Results: Dyspnea score decreased along the three measured days. Diaphragm and scalene EMG index showed a positive and significant direct relationship with dyspnea score (p<0.001 and p = 0.003 respectively) whereas pectoralis minor muscle did not. Conclusion: In our pilot study, diaphragm and scalene EMG activity was associated with increasing severity of dyspnea. Surface respiratory EMG could be a useful objective tool to improve assessment of dyspnea in acute HF patients.

for authors [JR] and research materials, but did not have any additional role in the study design, data collection and analysis, decision to publish, or preparation of the manuscript. The specific roles of this author are articulated in the 'author contributions' section.

**Competing interests:** I have read the journal's policy and the authors of this manuscript have the following competing interests: Jordi Rigau is employed by Sibel S.A.U. Additionally, Sibel S.A.U. kindly lent the cardio-respiratory polygraph for the study. This commercial affiliation does not alter our adherence to PLOS ONE policies on sharing data and materials. There are no patents, products in development or marketed products associated with this research to declare.

## Introduction

Dyspnea is the most common symptom in patients with acute heart failure (HF) and, as such, relief of breathlessness has been frequently employed as an end-point in clinical studies [1]. However, using dyspnea as an outcome variable has several limitations [1,2]. Indeed, there is no validated dyspnea scale in patients with HF [3] and the use of different dyspnea question-naires has not been standardized across studies. Moreover, previous research shows that it may be difficult to capture a meaningful change in dyspnea with the current scales available, which is consistent with the finding that dyspnea questionnaires are not interchangeable [4,5]. Furthermore, it is worth noting that the most frequently used dyspnea scales in HF do not take into account the psychological aspects of dyspnea or its sensory quality [6]. As a result of the limitations of dyspnea indices in HF, many studies focusing on dyspnea relief as end-point have produced conflicting results [2,3]. It is noteworthy that besides dyspnea questionnaires (which reflect the subjective patient sensation), there are no measurable physiological variables providing objective assessment of dyspnea in a setting of acute HF patients. Therefore, improving the tools we have to better capture the breathlessness sensation experienced by patients with acute HF would result in better characterizing patient progress and treatment.

Given that acute HF is associated with marked breathing dysfunction caused by diaphragm weakness [7], assessing the functional activity of respiratory muscles might provide useful information on the degree of dyspnea severity in HF patients. Indeed, it was early reported that HF patients had abnormally low values of maximum inspiratory pressure and impaired diaphragm contractibility and that these diaphragm alterations correlated with dyspnea perception and were associated with worse overall prognosis [8–10]. The concept that surface electromyography (EMG) of respiratory muscles correlates with the intensity of dyspnea has been already proved in patients with no HF but also presenting dyspnea as predominant symptom: mechanical ventilation [11,12] and chronic obstructive pulmonary disease (COPD) [13,14]. Accordingly, it is plausible to expect that similar results could be observed in HF patients. However, whether respiratory muscles EMG is correlated with dyspnea in acute HF patients is unknown. Thus, the aim of this study was to test the hypothesis that, in a clinical setting of acute HF, surface EMG of the patient's respiratory muscles correlates with the dyspnea scores conventionally assessed by using a clinical scale, potentially providing a non-invasive objective index helping to monitor patient breathlessness.

## Methods

### Patients

This prospective and descriptive pilot study was carried out in adult patients (>18 year old) admitted for acute HF at the Cardiology Department of the Hospital del Mar between August 2017 and September 2018. HF diagnosis and treatment was established according to the ESC HF Guidelines [15] and the investigation conformed to the principles outlined in the Declaration of Helsinki. Patients were included within the first 24 hours of admission and after obtaining their written informed consent to participate in the study, which was approved by the Hospital del Mar Ethical Committee (number 2017/7161/I). Patients with the following conditions were excluded: need for invasive or non-invasive mechanical ventilation, need for inotropic drugs, pacemaker-dependent, presence of other causes of acute dyspnea (e.g. decompensated COPD, pneumonia), metastatic active neoplasm or impossibility to give informed consent (e.g. low level of consciousness, moderate/severe cognitive impairment, and important language barrier).

## Protocol

Demographic and anthropometric data were collected. A first measurement was carried out within 24 h after hospital arrival. Clinical data (blood pressure, heart rate, spontaneous breath rate, oxygen saturation, weight) were recorded. To perform the EMG measurement, the patient was instrumented with a cardiorespiratory portable polygraph (Sleep&Go, Sibelmed, Spain) including nasal prongs to indirectly assess ventilation, pulse oximetry and surface EMG. To measure the activity of the main (diaphragm) and accessory (scalene and pectoralis minor) respiratory muscles we used 6 surface electrodes (Neuroline 715, Ambu, Copenhagen), a pair for each muscle (Fig 1). A ground electrode (Neuroline Ground, Ambu, Copenhagen) was placed on the right arm. All surface electrodes were placed on the patient's right side. The scalene electrode pair was placed in the posterior triangle of the neck between the sternocleido-mastoid muscle and the clavicle, the pectoralis minor electrode pair was placed in the second intercostal space, close to the sternum and the diaphragm electrode pair was placed in the seventh or eighth intercostal space at the mid-clavicular line (Fig 1). After the sensors were in

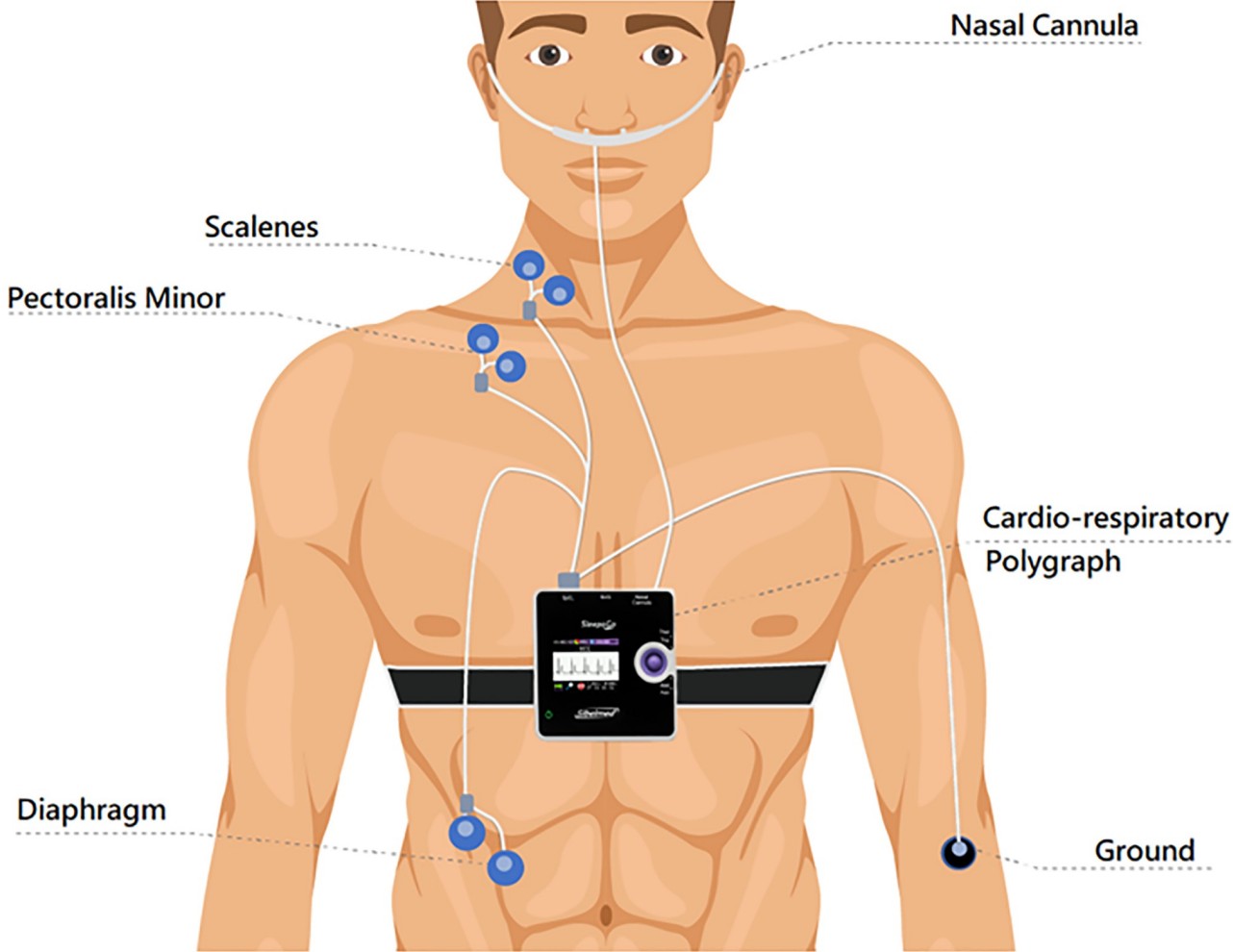

**Fig 1. Diagram of the measurement setting in heart failure patients.** Noninvasive electromyography (EMG) of respiratory muscles was assessed by surface electrodes. EMG signals and breathing flow indirectly sensed by a nasal cannula were recorded by a portable respiratory polygraph for subsequent data processing.

place, the patient was allowed to relax and stabilize his/her breathing sensation for a few minutes in sitting or semi-recumbent posture and was asked about his/her dyspnea sensation by means of the conventional Likert 5 questionnaire (score 1 = not at all short of breath, score 2 = mildly short of breath, score 3 = moderately short of breath, score 4 = severely short of breath, score 5 = worst possible short of breath). Then, the patient was instructed not to move or talk, and sensors data recording was started. After 3 min of spontaneous breathing the patient was asked to breathe at maximum effort for several cycles, data recording was subsequently stopped, and the measurement finished after the patient was freed from the EMG electrodes. The same measurement procedure was repeated along hospital admission at day 2 and day 5 or the day of discharge, whichever came first.

## EMG data processing

The clinicians performing the measurements on patients were blinded to the EMG data, which were analyzed offline following patient discharge. The EMG signals downloaded from the polygraph (Fig 2) were first processed to detect and remove the QRS complex. To this end, the position of the R peak in the QRS complex was detected by using the specialized Pan-

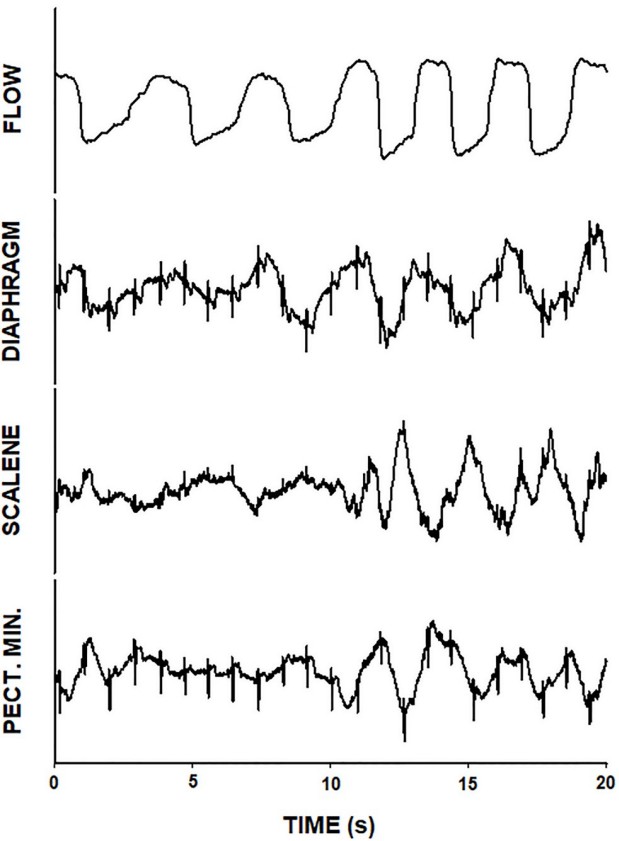

**Fig 2. Example of the EMG signals from the polygraph recorder during a normalizing breathing maneuver.**
Breathing flow and EMG of diaphragm, scalene and pectoralis minor. All these signals are measured in Volts in arbitrary scale since they correspond to an uncalibrated flow signal sensed by nasal prongs and to the muscle activity EMG signals with an amplitude that depends on the amplifier gain. The first cycles correspond to spontaneous breathing and the last ones (starting at time 10 s approx.) correspond to maximum effort breathing. Increase in flow amplitude was associated with augmented respiratory muscles activity.

Tompkins algorithm [16], and the QRS component was removed and replaced by the vicinity moving average values of the signal. Subsequently, the signal was rectified to compute the mean amplitude of the EMG signal which is the most commonly used single variable to capture respiratory muscles activity in similar settings [11, 12]. Then, an index to quantify the activity of each respiratory muscle and measurement time was computed from the amplitude of the EMG data. This process was simultaneously applied to the EMG signals recorded in the 3 muscles, for periods including at least 3 breathing cycles during both spontaneous breathing and maximal breathing effort. To limit the major sources of variability in the amplitude of EMG signals -which is mainly caused by differences in the exact place of electrode position and in the electrical impedance between the muscle and the electrode wire- we defined an intra-measurement normalized index. To this end, the mean amplitude of EMG signal during spontaneous breathing was divided by the mean amplitude of EMG signal during maximal breathing effort [17]. This index was expected to be low under normal breathing conditions (low muscle activity in spontaneous breathing as compared with high muscle activity during maximum breathing) and increase as breathing muscles were compromised (either because high activity during spontaneous breathing as compared with maximal breathing, or because low amplitude during maximal effort when muscles approach fatigue). Hence, a higher EMG index was expected as dyspnea intensity increased.

## Statistical analysis

Categorical variables are presented in absolute number and percentage and were analyzed with the Chi-Squared test. Continuous variables are expressed as mean ± standard deviation or median and 25–75% percentiles and were analyzed using ANOVA repeated measurements test, or Kruskal-Wallis test in case that the Shapiro-Wilk test indicated that variables were not normally distributed. Relationship between variables was measured by the coefficient of linear correlation. Assessment of changes in dyspnea score along days and muscle activity index as a function of score in dyspnea was carried out by Kruskal Wallis test and ANOVA trend analysis, respectively. Statistical analysis was performed with Stata/IC version 15.1. Statistical significance was considered for $p < 0.05$.

## Results

The study was carried out on 25 patients (44% women) who, on average, were 79 years old and presented a mean ejection fraction of 46%, with preserved ejection fraction (EF >50%) in 45% of them. Table 1 describes the main anthropometric and clinical patient data.

The first, second and third EMG measurements were carried out at 18 ± 5 h, 43 ± 7 h and 112 ± 22 h after admission into the Emergency Department, respectively. Table 2 shows the evolution of clinical and analytical parameters along the whole measurement period. As expected from patients successfully treated for an acute HF episode, we observed a decrease in respiratory and heart rate and, at day 5, only one third of patients needed oxygen supplementation. Renal function was stable and there was a slight increase in blood sodium, reflecting good response to treatment and decrease in volume overload. Although not measured in all patients, NT-proBNP decreased by more than 30% reflecting good response to therapy and improved prognosis. Consistently, dyspnea score decreased along the three measured days: from a median 2 (1–3) in day 1 to a median 1 (1–2) in day 5 (p = 0.021). Noteworthy, respiratory rate did not show a significant relationship with dyspnea score (p = 0.401).

Out of the total 75 possible measurements per muscle (25 patients x 3 days/patient), reliable EMG results were obtained in 66 cases (88%) for diaphragm and pectoralis minor and in 55 cases (73%) in scalene. The activity of the scalene (but not of the pectoralis minor) as measured

**Table 1. Baseline clinical and demographic characteristics of heart failure patients.**

| Variables | Value |
|---|---|
| *N* | 25 |
| Women | 11 (44) |
| Age, years | 79 ± 10 |
| Active smoker | 3 (12) |
| Heart failure within 12 months | 8 (32) |
| Heart failure within 30 days | 4 (29) |
| Ejection fraction, % | 46 ± 14 |
| Reduced LVEF (LVEF <50%) | 12 (55) |
| Atrial fibrillation | 16 (64) |
| Coronary artery disease | 10 (40) |
| Moderate to severe valve heart disease | 14 (56) |
| Chronic kidney disease | 18 (75) |
| Anemia | 14 (58) |
| Medication | |
| Beta-blocker | 21 (84) |
| ACE inhibitor-ARB/MRA/ARNI | 12 (48) |
| Mean oral furosemide dose, mg | 80 ± 40 |
| Hydrochlorothiazide | 4 (16) |

Data are n (%) or mean ± standard deviation. Anemia was defined as a hemoglobin < 13 g/dL in men and < 12 mg/dL in women. Chronic kidney disease was defined as an estimated Glomerular Filtration Rate (eGFR) < 60 mL/min/1.73m$^2$. ACE = Angiotensin Converting Enzyme; ARB = Angiotensin Receptor Blocker; MRA = Mineralocorticoid Receptor Antagonist; ARNI = Angiotensin Receptor Neprilysin Inhibitor.

**Table 2. Changes in clinical and analytical parameters along measurements.**

| Variables | First measurement | Second measurement | Third measurement | p |
|---|---|---|---|---|
| Routine observations | | | | |
| SBP, mmHg | 119 ± 17 | 122 ± 22 | 117 ± 21 | 0.636 |
| DBP, mmHg | 65 ± 12 | 62 ± 14 | 64 ± 15 | 0.916 |
| Heart rate, bpm | 81 ± 17 | 75 ± 14 | 72 ± 11 | 0.005 |
| Respiratory rate (RR), bpm | 25 ± 6 | 20 ± 7 | 18 ± 5 | <0.001 |
| Tachypnea (RR >25 bpm) | 10 (40) | 4 (17) | 4 (17) | 0.089 |
| Oxygen saturation, % | 95 ± 3 | 95 ± 2 | 95 ± 3 | 0.903 |
| Oxygen therapy | 20 (80) | 15 (60) | 8 (32) | <0.001 |
| Blood parameters | | | | |
| Creatinine, mg/dL | 1.27 ± 0.41 | 1.25 ± 0.38 | 1.25 ± 0.38 | 0.900 |
| Urea, mg/dL | 66 ± 31 | 75 ± 31 | 71 ± 24 | 0.255 |
| eGFR, mL/min/1.73m$^2$ | 53 ± 20 | 53 ± 19 | 53 ± 19 | 0.677 |
| Sodium, mEq/L | 139.9 ± 3.0 | 141.0 ± 3.6 | 141.4 ± 3.6 | 0.039 |
| Potassium, mEq/L | 4.14 ± 0.48 | 3.75 ± 0.39 | 4.13 ± 0.39 | 0.981 |
| NT-proBNP, pg/mL* | 7396 (3367–15066) | 3853 (1616–8423) | 3333 (1534–5038) | 0.002 |

Data are n (%), mean ± standard deviation or median (interquartile range). SBP = Systolic Blood Pressure; DBP = Diastolic Blood Pressure; eGFR = estimated Glomerular Filtration Rate.

* NT-proBNP was only measured in 12 patients at Day 2 and 19 patients at Day 5.

by surface EMG was significantly correlated with that of the diaphragm (r = 0.53; p < 0.001). No significant correlation was observed between the breathing frequency and the diaphragm EMG activity index. The scalene and particularly the diaphragm (but not the pectoralis minor) activity measured by the EMG index showed a significant direct relationship with the clinical dyspnea index. Indeed, the coefficients of correlation between Likert 5 and scalene and diaphragm EMG were 0.45 (p = 0.002; F = 9.15) and 0.48 (p < 0.001; F = 14.07), respectively. As illustrated by Fig 3, the higher the dyspnea index the higher EMG index (p < 0.001 and

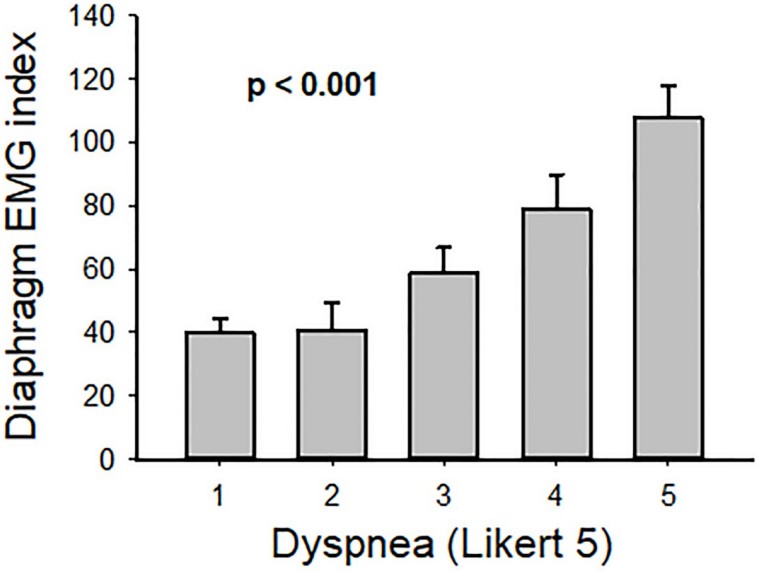

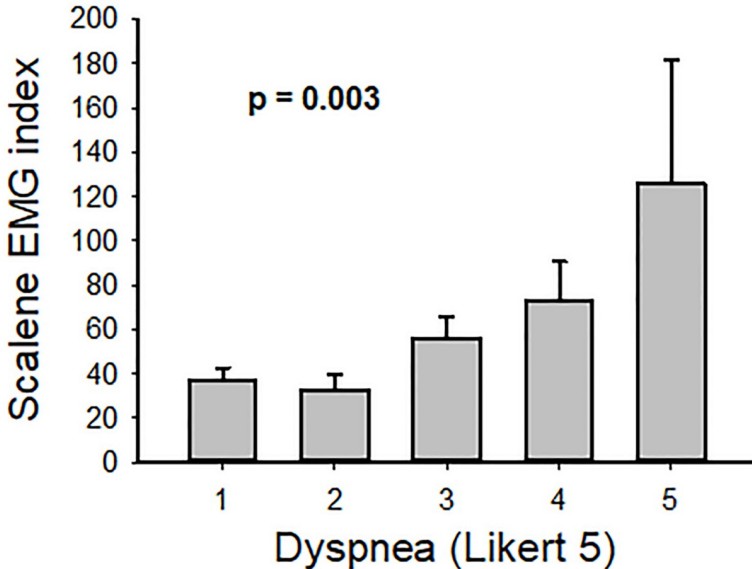

**Fig 3. Diaphragm and scalene EMG activity index as a function of dyspnea (Likert 5 scale).** EMG indices increased with dyspnea severity. Data are EMG index for measurements (all patients, all days) with a given dyspnea index value. Data are mean ± SE. See text for more details.

p = 0.003 respectively). By contrast, the pectoralis minor muscle did not show a significant relationship with the clinical dyspnea scale.

## Discussion

To the best of our knowledge this is the first study that has analyzed the relationship between surface respiratory electromyography (EMG) and dyspnea in acute HF patients. This pilot study strongly suggests that non-invasively measuring the activity of respiratory muscles is feasible in the clinical setting of acute HF patients and that increased activity of diaphragm and scalene is significantly associated with the severity of dyspnea measured by a conventional clinical questionnaire.

Surface EMG has some limitations as compared with invasive EMG based on needle electrodes directly sensing the muscle tissue. One limitation is that EMG signal amplitude depends on the electrical quality of the electrode attachment on the skin. Intra-patient variability in this interface impedance affects the EMG amplitude when repeating measurements in different days with newly placed electrodes in a given patient. Inter-patient variability may result from positioning the electrodes at slightly different positions and also by variation in the impedance from the muscle to the skin, for instance because of different amount of subcutaneous fat tissue. This type of intra- and inter-variability can be reduced by computing an EMG index including an intra-measurement normalization from maximal breathing maneuvers. Another limitation of surface EMG is that although the electrodes are placed on skin zones close to the muscle of interest to mainly capture the electric potentials from that muscle, some signal contamination by electrical activity from other muscles is possible, specifically regarding respiratory muscles. However, recent studies have provided validation data by comparing surface respiratory EMG with esophageal diaphragm EMG [18] and by using surface EMG to assess inspiratory effort [19]. Notwithstanding its limitations, surface EMG is a useful technique for clinical purposes given its non-invasiveness.

Dyspnea is defined as "a subjective experience of breathing discomfort that consists of qualitatively distinct sensations that vary in intensity" [20] and is the most common symptom in patients with acute HF. Thus, dyspnea relief has been frequently used as an end-point in clinical studies [1], but limitations should be considered [1,2]. On the one hand, there is not a validated dyspnea scale in patients with HF [3]. A consensus group recommended the use of the same instrument at baseline and repeatedly thereafter over just asking the patient whether symptoms have changed or not [3]. On the other hand, a limitation of the use of the dyspnea scales in HF is the lack of standardization across studies. Although a proposal to standardize dyspnea measurement in acute HF has been published [3], studies in acute HF use different scales at different time points and how dyspnea was measured is frequently not described. Some studies assess dyspnea as early as within 1 hour of first medical evaluation [21] whereas others could include patients within 48 hours of hospitalization [22]. Our first measurement was carried out at 18 ± 5 h from admission into the Emergency Department, therefore capturing the worst possible clinical scenario. Interestingly, the severity of dyspnea according to the dyspnea scale was not severe (a median of 2 with Likert 5 scale (i.e. mildly short of breath) even in the most acute phase. However, all patients were in need of oxygen therapy and respiratory rate was high (Table 2). These baseline characteristics were similar to other studies in acute HF [4]. In previous reports, only very severely baseline dyspnea was associated with dyspnea improvement by 5-point Likert Scale whereas moderately and severe dyspnea were not. In contrast, baseline dyspnea was associated with dyspnea improvement by Visual Analog Scale [4]. These data show that the most frequently used dyspnea scales are not interchangeable. Furthermore, not all scales have an established minimal clinically important difference

[23] and it may be difficult to capture a meaningful change in dyspnea with the current scales available [4,5]. Finally, psychological and emotional distress has been significantly associated with dyspnea in HF [24,25] but these components or alterations of sensorial quality are usually nor assessed in the most commonly scales used for dyspnea assessment [6]. Accordingly, it is not surprising that many HF drug studies that have dyspnea relief as end points have failed to show a difference between the active drug and placebo. Whether they reflect a real lack of effect of the medication tested or the use of a suboptimal tool to assess the endpoint is unclear. Given the shortcomings of common dyspnea scales and their subjective nature, from a clinical point of view it would be useful to have an objective index that would bring additional information about dyspnea.

Since acute HF is associated with marked inspiratory dysfunction [7–10], we hypothesized that assessing the activity of respiratory muscles could provide a useful index associated with dyspnea in these patients. In addition to the diaphragm, which is the main inspiratory muscle, we also focused on accessory respiratory muscles since they are recruited in healthy subjects under conditions more strenuous than spontaneous breathing at rest [26,27] as well as in patients with respiratory diseases presenting dyspnea as a prominent symptom. For example, in patients with COPD, a disease frequently associated with dyspnea both in the stable and acute phase, surface inspiratory electromyograms (EMG) of respiratory muscles have been correlated with the intensity of dyspnea and prognosis [13,14]. Even when treatment with salmeterol-fluticasone in severe COPD was not associated with significant change in hyperinflation or pulmonary mechanics, this treatment induced a significant decrease in activity of the chest wall parasternal inspiratory muscle [28], suggesting that EMG could be a sensitive measure to monitor improvement in these respiratory patients. Surface EMG also correlated with breathlessness in patients with cystic fibrosis during exercise [29]. Another example showing a relationship between dyspnea and respiratory muscles activity is mechanical ventilation. In this setting, a strong correlation between accessory respiratory EMG activity and the severity of dyspnea was described in patients subjected to invasive pressure support ventilation [11]. It has also been reported that scalene EMG activity level was correlated with dyspnea intensity in subjects under non-invasive mechanical ventilation [12,30]. The results obtained in the present study indicate that respiratory muscles EMG correlate with dyspnea not only in patients in whom breathlessness is induced by a respiratory system disease, but also in acute HF, a situation where the main cause of dyspnea is not a respiratory disorder. In particular, in our HF patients we observed a correlation between diaphragm and scalene EMGs, the two respiratory muscles that showed significant correlation with dyspnea (Fig 3). Interestingly, the lack of correlation observed between respiratory frequency and diaphragm EMG is consistent with our finding that breathing frequency did not depend on dyspnea score and with the fact that the breathing rate has not been previously reported as a good index to assess dyspnea.

Dyspnea, a subjective sensation experienced by patients with acute HF, is determined by both physical and psychological components which are still poorly understood. Although alteration of breathing activity is clearly accompanying the dyspneic events in HF patients, we did not expect that surface EMG of respiratory muscles provided a full assessment of dyspnea. However, the correlation we found between Likert 5 scale and diaphragm activity suggests that surface EMG could play a role as an objective tool helping to better assess dyspnea in acute HF. Interestingly, the measurement of surface EMG provided a feasible parameter with up to 88% technically correct measures overall. Notwithstanding the fact that we carried out single measurements, estimation of the EMG index could be carried out repeatedly and sequentially at different time intervals to both increase estimation robustness and provide time course monitoring. In this connection, future clinical applications could be developed for automatically monitoring patient's dyspnea, in particular at the patient's home (with potential

telemetric follow-up). In fact, small, portable and relatively cheap recorders including surface EMG are currently available for home monitoring sleep apnea, and new miniaturized and wireless EMG devices are being developed [31–33].

This proof of concept pilot study has some limitations. It only included patients with breathlessness at inclusion but many patients with acute HF are comfortable at rest but breathless on slight exertion. However, these patients, who can account for as much as 56% of acute HF patients, have usually also been excluded from clinical trials [34], especially those with dyspnea improvement as an end-point. Given our small sample size, we did not attempt to assess whether EMG values can be associated with prognosis. Moreover, dyspnea was assessed by one of the most common clinical questionnaires (Likert 5). As it has been described that other usual clinical indices (such as Visual Analog Scale) are not interchangeable to assess acute HF, future studies should assess whether surface respiratory EMG captures dyspnea as compared with other clinical indices of dyspnea. The results from this pilot study also suggest to further investigating the correlation between dyspnea severity and diaphragm EMG index over time adjusting for the change in NT-pro BNP and other biomarkers. Such a mechanistic research would allow to unravel the physiological signal captured by the EMG index and to ascertain whether diaphragm involvement in HF failure is a mere consequence of hypoperfusion and/or volume overload or is mediated by HF-related proinflammatory cytokines upregulation thereby independently contributing to dyspnea in these patients. This mechanistic information would help to optimally target the patient phenotype and clinical status for potential clinical application of surface EMG to assess dyspnea in HF.

## Conclusions

Non-invasive measurement of respiratory muscles EMG is feasible in the setting of acute HF patients. Diaphragm, and less strongly scalene, EMG index showed a significant direct relationship with Likert 5 dyspnea scale. The results in this study provide a proof of concept that surface EMG index could be used as a marker of breathlessness severity and employed as a more objective end-point in acute HF trials in clinical settings as well as a potential home monitoring tool in HF patients at risk. However, more research and future clinical trials are required to substantiate the novel results obtained in this pilot study.

## Author Contributions

**Conceptualization:** Ramon Farré, Núria Farré.

**Formal analysis:** Daniele Luiso, Jair A. Villanueva, Ramon Farré, Jordi Rigau, Núria Farré.

**Funding acquisition:** Ramon Farré, Núria Farré.

**Investigation:** Daniele Luiso, Jair A. Villanueva, Laia C. Belarte-Tornero, Aleix Fort, Zorba Blázquez-Bermejo, Sonia Ruiz, Julio Martí-Almor.

**Methodology:** Jordi Rigau, Núria Farré.

**Project administration:** Núria Farré.

**Supervision:** Núria Farré.

**Writing – original draft:** Daniele Luiso, Ramon Farré, Núria Farré.

**Writing – review & editing:** Daniele Luiso, Jair A. Villanueva, Laia C. Belarte-Tornero, Aleix Fort, Zorba Blázquez-Bermejo, Sonia Ruiz, Ramon Farré, Jordi Rigau, Julio Martí-Almor, Núria Farré.

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
