## [Decision Letter · Decision Letter 0]

19 Feb 2020

PONE-D-20-02315

Surface respiratory electromyography and dyspnea in acute heart failure patients.

PLOS ONE

Dear Dr Farré,

Thank you for submitting your manuscript to PLOS ONE. After careful consideration, we feel that it has merit but does not fully meet PLOS ONE’s publication criteria as it currently stands. Therefore, we invite you to submit a revised version of the manuscript that addresses the points raised during the review process.

We would appreciate receiving your revised manuscript by Apr 04 2020 11:59PM. To enhance the reproducibility of your results, we recommend that if applicable you deposit your laboratory protocols in protocols.io, where a protocol can be assigned its own identifier (DOI) such that it can be cited independently in the future. For instructions see: http://journals.plos.org/plosone/s/submission-guidelines#loc-laboratory-protocols

We look forward to receiving your revised manuscript.

Kind regards,

Nizam Uddin Ahamed, PhD

Academic Editor

PLOS ONE

Journal Requirements:

1. Please provide in a Table the data underlying Figure 2

2. Thank you for including your competing interests statement; "I have read the journal's policy and the authors of this manuscript have the following competing interests: Jordi Rigau is employed by Sibel S.A.U."

We note that one or more authors are affiliated with the commercial organisation; 

Sibel SA,U

Reviewers' comments:

Reviewer's Responses to Questions

**Comments to the Author**

1. Is the manuscript technically sound, and do the data support the conclusions?

Reviewer #1: Partly

Reviewer #2: Yes

Reviewer #3: Partly

Reviewer #4: Yes

2. Has the statistical analysis been performed appropriately and rigorously? 

Reviewer #1: Yes

Reviewer #2: No

Reviewer #3: No

Reviewer #4: Yes

3. Have the authors made all data underlying the findings in their manuscript fully available?

Reviewer #1: No

Reviewer #2: Yes

Reviewer #3: Yes

Reviewer #4: Yes

4. Is the manuscript presented in an intelligible fashion and written in standard English?

Reviewer #1: No

Reviewer #2: Yes

Reviewer #3: Yes

Reviewer #4: Yes

5. Review Comments to the Author

Reviewer #1: The authors present a method to assess dyspnea. In my opinion, the manuscript could be useful in practice but my main concern is that it does not contain important information relating to methodology. I found that the manuscript fails to make a significant contribution beyond previously reported work (the manuscript just uses the different index to measure dyspnea compared to previous works).

The organization of the work is not quite good and clear. I would like to convey a few shortcomings and comments that I have seen in the manuscript:

- Some affiliations are not in English.

- It is not mentioned where the data can be found.

- The introduction section needs to be revised.

- EMG data processing is not described clearly. For example:

- How the QRS was detected and removed?

- How did the authors identify the activity of each respiratory muscle?

Reviewer #2: The present study provides a proof of concept for monitoring of dyspnea condition by correlating EMG activity in respiratory muscles with the Likert 5 questionnaire for dyspnea severity. The idea sounds interesting and the paper is well-written, however, before being considered for publication some limitations need to be addressed by the authors:

1) Abstract: authors claim that there is no instrument for objective evaluation of dyspnea, however, I think it may not be a 100% correct statement, since I could find at least one possible device to be used for this purpose through respiratory measurements. See below:

https://breathe.ersjournals.com/content/breathe/1/2/100.full.pdf

Authors should be careful with such claims.

2) Protocol: The authors provided detailed descriptions for the electrode placements, however, a visual figure could help in making the process much more easy to follow.

3) EMG data processing:

- Line 122: Please explain the QRS complex and the procedure of removing it briefly.

- Line 123: Why amplitude is the only feature considered here? Why not other signal-related features (e.g. peak, frequency of peaks, etc.) were not considered. Please explain.

4) Statistical Analysis:

- How did you decide on these tests? Did you have any assumption on the distribution of data? If yes, why do you use a non-parametric method (Kruskal Wallis test)? Please explain.

A minor comment:

Line 84: "according the ESC ..." should be "according to the ESC ..."

Reviewer #3: The authors propose to use surface EMG of respiratory muscles as an objective way to detect and monitor short of breath for acute heart failure patients. To this end, they wanted to test the hypothesis that the severity of dyspnea is correlated to an EMG index (mean magnitude normalized by maximum muscle contraction) of respiratory muscles. They designed a procedure and collected data from 25 patients and post processed EMG data. They show that the severity of dyspnea is positively correlated with the magnitude of the EMG and concluded that this could be used as a clinical procedure.

The authors adopted similar approaches studied in the literature. The novelty of the paper is in its application with HF patients. The organization of the paper is good. The reviewer has a few comments as following:

(1) Could the authors include the total time needed for the procedure? Including sensor placement, skin preparation, etc.

(2) The interpretation of table 2 is not clear. The authors included many clinical measurements without explaining them.

(3) Speaking of table 2, it seems the respiratory rate would correlate well with both Likert and EMG. So why not use this variable to indicate short of breath instead of using EMG? After all, as the authors mention, EMG has so many limitations in practice.

(4) One of the disadvantages of surface EMG that the authors did not mention is that it registers heartbeat, as shown in figure 1. Did authors apply filters to remove/minimize this effect? Was this the reason why the authors placed all electrodes on the right side of the body?

(5) The statistical analysis is not complete. For linear correlation, please report the method used, normality of variables, residuals, normality of residuals, etc. In addition to P value, please also report F value.

(6) It's unclear how the technology will be used in clinic. One of the advantages to use EMG over Likert, according to the authors, is that it doesn't require patient cooperation. This is a little bit confusing because the procedure requires the patient to breathe at maximum effort for several cycles.

Reviewer #4: This paper presents a study on diaphragm and scalene surface EMG signals in order to evaluate their feasibility to improve assessment of dyspnea in acute heart failure patients.

The paper is quite well written and organized, and the reported findings are interesting.

However, the reported results must be better displayed in fig. 1 (e.g. values on y axes are not displayed for both air flow and SEMG signals)

6. PLOS authors have the option to publish the peer review history of their article (what does this mean?). If published, this will include your full peer review and any attached files.

Reviewer #1: No

Reviewer #2: No

Reviewer #3: No

Reviewer #4: No

---

## [Author Response · Author response to Decision Letter 0]

24 Mar 2020

We have adapted the style to the journal requirements and modified the Funding Statement so now it reads: “The funder provided support in the form of salaries for authors [JR] and research materials, but did not have any additional role in the study design, data collection and analysis, decision to publish, or preparation of the manuscript. The specific roles of these authors are articulated in the ‘author contributions’ section”. Finally, we have modified the Competing Interests Statement: “This commercial affiliation does not alter our adherence to PLOS ONE policies on sharing data and materials”.

(Please note that, in our answers, line numbers indicating changes correspond to lines in the colored marked-up version of the revised manuscript.) 

REVIEWER #1: 

The authors present a method to assess dyspnea. In my opinion, the manuscript could be useful in practice but my main concern is that it does not contain important information relating to methodology. I found that the manuscript fails to make a significant contribution beyond previously reported work (the manuscript just uses the different index to measure dyspnea compared to previous works).

The organization of the work is not quite good and clear. I would like to convey a few shortcomings and comments that I have seen in the manuscript:

ANSWER: Thanks for considering that the manuscript could be useful in practice. We agree with the reviewer in that our work does not provide innovative methodology. Indeed, we have used the methods previously employed to assess surface EMG of respiratory muscles and dyspnea in patients with respiratory diseases. However, the aim novelty of our study is that we tested the hypothesis that this noninvasive tool provides an index related to dyspnea in the particular, and very clinically relevant population, of patients with heart failure. 

- Some affiliations are not in English.

ANSWER: In the revised version all affiliations are in English

- It is not mentioned where the data can be found.

ANSWER: Thanks for noting that we forgot to provide this section:

Data availability: All the relevant data are provided within the manuscript. The participant patients did not consent to having details of their data publicly available. However, requests for access to some specific data may be directed to the email address of the corresponding author (NFarreLopez@parcdesalutmar.cat).

- The introduction section needs to be revised.

ANSWER: Following the Reviewer suggestion, the Introduction section, particularly the second paragraph, has been revised.

- EMG data processing is not described clearly. For example:

- How the QRS was detected and removed?

- How did the authors identify the activity of each respiratory muscle?

ANSWER: The data processing to remove the QRS noise from the signals has been explained in more detail in the revised manuscript (lines 143-146). We have also explained the signal periods used for computing EMG activity indices (lines 146-152). 

REVIEWER #2: 

The present study provides a proof of concept for monitoring of dyspnea condition by correlating EMG activity in respiratory muscles with the Likert 5 questionnaire for dyspnea severity. The idea sounds interesting and the paper is well-written, however, before being considered for publication some limitations need to be addressed by the authors:

ANSWER: Thank you for your positive comments. We address the specific point you raised:

1) Abstract: authors claim that there is no instrument for objective evaluation of dyspnea, however, I think it may not be a 100% correct statement, since I could find at least one possible device to be used for this purpose through respiratory measurements. See below:

https://breathe.ersjournals.com/content/breathe/1/2/100.full.pdf

Authors should be careful with such claims.

ANSWER: We agree with the Reviewer in that the short sentence in the abstract of the original manuscript may induce confusion. We have modified it (lines 26-30) to be more specific by stating that besides dyspnea questionnaires (which reflect the subjective patient sensation and are not fully validated in HF) there are no measurable physiological variables providing objective assessment of dyspnea in HF patients.

2) Protocol: The authors provided detailed descriptions for the electrode placements, however, a visual figure could help in making the process much more easy to follow.

ANSWER: As suggested by the Reviewer, a new Figure 1 has been added in the revised manuscript.

3) EMG data processing:

- Line 122: Please explain the QRS complex and the procedure of removing it briefly.

ANSWER: The data processing to remove the QRS noise from the signals has been explained in more detail in the revised manuscript (lines 143-146).

- Line 123: Why amplitude is the only feature considered here? Why not other signal-related features (e.g. peak, frequency of peaks, etc.) were not considered. Please explain.

ANSWER: We used the EMG amplitude index since this is the most commonly used single variable to capture respiratory muscles activity in similar settings (for instance in references 11 and 12). We have mentioned it in the revised manuscript (lines 146-152).

4) Statistical Analysis:

- How did you decide on these tests? Did you have any assumption on the distribution of data? If yes, why do you use a non-parametric method (Kruskal Wallis test)? Please explain.

ANSWER: We have provided the explanation in the revised manuscript (lines 177-178).

A minor comment:

Line 84: "according the ESC ..." should be "according to the ESC ..."

ANSWER: Done.

REVIEWER #3: 

The authors propose to use surface EMG of respiratory muscles as an objective way to detect and monitor short of breath for acute heart failure patients. To this end, they wanted to test the hypothesis that the severity of dyspnea is correlated to an EMG index (mean magnitude normalized by maximum muscle contraction) of respiratory muscles. They designed a procedure and collected data from 25 patients and post processed EMG data. They show that the severity of dyspnea is positively correlated with the magnitude of the EMG and concluded that this could be used as a clinical procedure.

The authors adopted similar approaches studied in the literature. The novelty of the paper is in its application with HF patients. The organization of the paper is good. 

ANSWER: Thanks for your positive comment.

The reviewer has a few comments as following:

(1) Could the authors include the total time needed for the procedure? Including sensor placement, skin preparation, etc.

ANSWER: It is difficult to provide a clear answer to this question, particularly in view of potential interest for future routine applications. On the one hand, the time required for each measurement considerably depended on the specific patient and on the level of dyspnea he/she was experiencing at the moment of the measurement. On the other hand, as this was a pilot (not a routine) study, it involved a learning curve in its practical implementation and thus in the time required. Moreover, in the pilot study reported here, 3 respiratory muscles were measured in each patent. Nevertheless, any potential future application in routine would involve just monitoring one muscle EMG (diaphragm or scalene), with substantial reduction in time, particularly if the process is carried out by a specifically trained professional. 

(2) The interpretation of table 2 is not clear. The authors included many clinical measurements without explaining them.

ANSWER: In the Results section we have now added a description of the changes observed along the study period in the most relevant clinical parameters shown in Table 2 (lines 203-208).

(3) Speaking of table 2, it seems the respiratory rate would correlate well with both Likert and EMG. So why not use this variable to indicate short of breath instead of using EMG? After all, as the authors mention, EMG has so many limitations in practice.

ANSWER: The breathing frequency is modified by the HF severity status, as it is well-known and as it is also reflected in Table 2 showing that breathing rate decreased along the days after admission for HF exacerbation. However, in agreement with previous studies, we found no significant relationship between breathing frequency and dyspnea score. In the revised manuscript we have reported this lack of relationship in Results (lines 209-210) and we have mentioned it in the Discussion (lines 323-324).

(4) One of the disadvantages of surface EMG that the authors did not mention is that it registers heartbeat, as shown in figure 1. Did authors apply filters to remove/minimize this effect? Was this the reason why the authors placed all electrodes on the right side of the body

ANSWER: In the original manuscript we very briefly mentioned that a filter for the heartbeat noise was applied. However, we have now explained the QRS filter used in more detail (lines 143-146). Yes, we placed the electrodes on the body right side to minimize the heartbeat noise into the surface EMG of respiratory muscles,.

(5) The statistical analysis is not complete. For linear correlation, please report the method used, normality of variables, residuals, normality of residuals, etc. In addition to P value, please also report F value.

ANSWER: Thank you for your comment. To assess the linear correlation between Likert 5 scale and EMG index we used an ANOVA trend analysis test (line 182): with a p-value less than 0.05, the null hypothesis that the variables were not linear correlated was rejected. In the revised version we included F value for correlation tests between Likert 5 scale and EMG indices (line 227).

(6) It's unclear how the technology will be used in clinic. One of the advantages to use EMG over Likert, according to the authors, is that it doesn't require patient cooperation. This is a little bit confusing because the procedure requires the patient to breathe at maximum effort for several cycles.

ANSWER: Thank you for your comment. Certainly, some cooperation is required from the patient. To avoid any confusion, the way we ended our final sentence in Introduction has been corrected (lines 89-90). 

REVIEWER #4: 

This paper presents a study on diaphragm and scalene surface EMG signals in order to evaluate their feasibility to improve assessment of dyspnea in acute heart failure patients.

The paper is quite well written and organized, and the reported findings are interesting.

ANSWER: Thanks for your positive comment.

H

owever, the reported results must be better displayed in fig. 1 (e.g. values on y axes are not displayed for both air flow and SEMG signals)

ANSWER: Thanks for noting it. In the legend of Figure 2 (former fig.1) we have now indicated: “All these are voltage signals in arbitrary scale since they correspond to an uncalibrated flow signal sensed by nasal prongs and to the muscle activity EMG signals with amplitude that depends

---

## [Decision Letter · Decision Letter 1]

10 Apr 2020

Surface respiratory electromyography and dyspnea in acute heart failure patients.

PONE-D-20-02315R1

Dear Dr. Farré,

We are pleased to inform you that your manuscript has been judged scientifically suitable for publication and will be formally accepted for publication once it complies with all outstanding technical requirements.

With kind regards,

Nizam Uddin Ahamed, PhD

Academic Editor

PLOS ONE

Additional Editor Comments (optional):

Reviewers' comments:

Reviewer's Responses to Questions

**Comments to the Author**

1. If the authors have adequately addressed your comments raised in a previous round of review and you feel that this manuscript is now acceptable for publication, you may indicate that here to bypass the “Comments to the Author” section, enter your conflict of interest statement in the “Confidential to Editor” section, and submit your "Accept" recommendation.

Reviewer #1: All comments have been addressed

Reviewer #2: All comments have been addressed

Reviewer #4: All comments have been addressed

2. Is the manuscript technically sound, and do the data support the conclusions?

Reviewer #1: Yes

Reviewer #2: Yes

Reviewer #4: (No Response)

3. Has the statistical analysis been performed appropriately and rigorously? 

Reviewer #1: Yes

Reviewer #2: Yes

Reviewer #4: (No Response)

4. Have the authors made all data underlying the findings in their manuscript fully available?

Reviewer #1: Yes

Reviewer #2: Yes

Reviewer #4: (No Response)

5. Is the manuscript presented in an intelligible fashion and written in standard English?

Reviewer #1: Yes

Reviewer #2: Yes

Reviewer #4: (No Response)

6. Review Comments to the Author

Reviewer #1: (No Response)

Reviewer #2: (No Response)

Reviewer #4: (No Response)

7. PLOS authors have the option to publish the peer review history of their article (what does this mean?). If published, this will include your full peer review and any attached files.

Reviewer #1: No

Reviewer #2: No

Reviewer #4: No

---

## [Editor Report · Acceptance letter]

17 Apr 2020

PONE-D-20-02315R1 

Surface respiratory electromyography and dyspnea in acute heart failure patients. 

Dear Dr. Farré:

I am pleased to inform you that your manuscript has been deemed suitable for publication in PLOS ONE. Congratulations! Your manuscript is now with our production department. 

With kind regards,

on behalf of

Dr. Nizam Uddin Ahamed 

Academic Editor

PLOS ONE